# UAD-CMPT: Unified Face Attack Detection via Cross-Modal Prompt Tuning

## Abstract

Pre-trained vision–language models (VLMs), such as CLIP, fail to realize their anticipated superiority in the Unified Face Attack Detection (UAD) task. We attribute this to two task-specific challenges: (1) Categorical ambiguity. UAD categories such as *live* and *fake* pose challenges for semantic alignment in CLIP, as they are subjectively defined concepts rather than literal meanings. (2) Forgery diversity. The diversity of forgery cues across physical and digital attacks hinders the language modality from delineating reliable decision boundaries. To address these issues, we propose Cross-Modal Prompt Tuning (**CMPT**), a bidirectional prompt-transfer framework that realigns vision and language. In the language branch, Synonym Semantic Augmentation (**SSA**) retrieves semantically related neighbors from a frozen vocabulary and integrates them via similarity-weighted aggregation, enriching category semantics and targeting comprehensive coverage of category expressions. In the vision branch, a Fourier-based High-Frequency Amplifier (**FHFA**) suppresses low frequencies and adaptively strengthens the real and imaginary components of high-frequency signals with learnable convolutions, consolidating diverse forgery cues into a shared discriminative space. Within UAD-CMPT, the resulting semantically augmented categories are sent to the vision branch, and instance-conditioned visual prompts encoding decision criteria are returned to the language branch; both act as learnable prompts to achieve vision–language alignment. Extensive experiments demonstrate that UAD-CMPT consistently outperforms state-of-the-art methods on multiple UAD benchmarks.

## 1 Introduction

A face recognition system encounters two threats: physical presentation attacks, such as printed photos Zhang et al. (2020b); Guo et al. (2022), video replays Boulkenafet et al. (2017), and 3D masks Liu et al. (2018a), which occur before the sensor captures the face; and digital deepfake attacks, including face swapping, attribute editing Yan et al. (2024), and face synthesis, which occur after capture. The former is addressed by Face Anti-Spoofing (FAS) Yu et al. (2020a); Zhang et al. (2020a); Zhou et al. (2022c). At the same time, the latter relies on DeepFake Detection (DFD) Bei et al. (2024); Yan et al. (2023); Li et al. (2024). These tasks are usually treated as separate problems, which inevitably increases the cost of model deployment and computation. However, since both physical and digital attacks originate from live faces through different forgery techniques, they share a common discriminative space that allows them to be categorized under a unified class. This motivates the Unified Attack Detection (UAD) task Deb et al. (2023); Fang et al. (2024); Liu et al. (2025), which highlights the possibility and importance of using a single model to jointly defend against diverse physical and digital forgeries.

Recently, vision–language models (VLMs) Radford et al. (2021) have demonstrated strong generalization on diverse downstream classification tasks Zhou et al. (2022a); Khattak et al. (2023); Gao et al. (2024), yet their performance on UAD Zou et al. (2024); Chen et al. (2025); Li et al. (2025a) remains unsatisfactory, largely due to two task-specific challenges: (1) categorical ambiguity and (2) forgery diversity. As illustrated in Fig. 1(a), UAD is cast as a binary problem in which the label *live* denotes genuine faces, whereas *fake* uniformly encompasses physical forgeries such as printed photos, video replays, and 3D masks, as well as digital forgeries such as face swapping, attribute editing, and face synthesis. However, as shown in Fig. 1(b), human-defined vision–language mappings are difficult for VLMs to capture, given their training on generic image–text pairs. Consequently, VLMs often ground

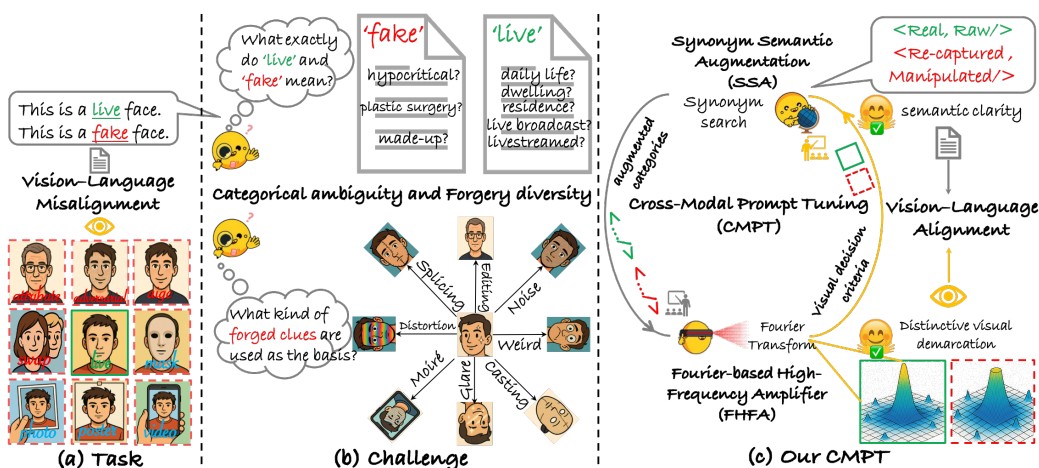

Figure 1: UAD under a vision-language framework. (a) It labels genuine samples as *live* and uniformly assigns diverse physical and digital forgeries to *fake*. (b) Categorical ambiguity: The semantics of human-defined textual labels (*live/fake*) are ambiguous. Forgery diversity: The diversity and heterogeneity of forgery cues make classification criteria difficult to articulate. (c) In UAD-CMPT, SSA (language→vision) generates semantically augmented category prompts; FHFA (vision→language) suppresses low frequencies and enhances high-frequency real/imaginary parts to consolidate forgery cues into a shared discriminative space; the resulting bidirectional prompts achieve alignment.

*live* in senses like "daily life", "residence", or "livestream", and interpret *fake* as "hypocritical", "plastic surgery", or "make-up". Meanwhile, forgery cues are highly diverse, producing heterogeneous visual characteristics that a single label *fake* cannot uniformly represent. Taken together, alleviating categorical ambiguity and strengthening the semantic commonality of forgery cues are pivotal to making VLMs effective for UAD.

Categorical ambiguity and the difficulty of inducing classification criteria disrupt the pretrained alignment of VLMs, necessitating UAD-specific realignment. Considering that re-establishing the mapping between categories and visual features is impractical, supplementing the vision modality with category prompts and the language modality with visual prompts offers a cost-effective alignment strategy. Accordingly, we require category prompts in the visual modality to possess two properties: (1) Semantic synonymy with categories. Expand each fixed textual category into task-aligned synonym descriptors covering all visual instances, e.g., expand *fake* to printed photo, video replay, and face editing. (2) Induce category-discriminative cues from all visual tokens. The prompt should interact with all visual tokens and summarize category-relevant cues; for example, color distortion in printed photos, screen moiré in video replays, and splicing artifacts in face swapping. For the visual prompt, we expect it to provide discriminative decision criteria by unifying heterogeneous forgery cues while remaining separable from genuine samples. This implies that the visual prompt originates from a discriminative visual feature space and is mapped into the language modality to assist categories in delineating decision boundaries.

As shown in Fig. 1(c), we first propose Cross-Modal Prompt Tuning (**CMPT**), a bidirectional prompt-transfer framework that restores vision–language alignment. Then, along the language-to-vision direction in UAD-CMPT, we introduce a Synonym Semantic Augmentation (**SSA**) module. It expands a fixed textual category into a task-aligned set of synonymized descriptors by retrieving semantically related neighbors from a frozen vocabulary and integrating them with similarity-weighted aggregation. The resulting semantically augmented category copies act as category prompts for the vision branch to induce category-relevant cues. Meanwhile, along the vision-to-language direction, we introduce a Fourier-based High-Frequency Amplifier (**FHFA**) that suppresses low frequencies and adaptively amplifies the real and imaginary parts of high-frequency signals with learnable convolutions. FHFA consolidates heterogeneous forgery cues into a shared discriminative space and produces instance-conditioned visual prompts separable from genuine samples, which are mapped to the language modality to assist category boundary delineation. Finally, SSA and FHFA instantiate UAD-CMPT's bidirectional prompt transfer to restore pretrained vision-language alignment in the UAD space.

## 2 RELATED WORK

**Face Anti-Spoofing (FAS).** FAS was initially designed to counter physical attacks such as printed photos, video replays, and 3D masks. Early CNN-based approaches Liu et al. (2018b); Yu et al. (2020b) achieved strong performance on seen domains but suffered sharp degradation under domain shifts, exposing poor domain generalization (DG). To address this, domain DG FAS methods Liu et al. (2024b); Cai et al. (2024); Hu et al. (2024); Wang et al. (2024) aim to remain effective on unseen domains. With the rise of multimodal models and contrastive learning, recent works demonstrate that textual descriptions can guide visual feature weighting and improve generalization Srivatsan et al. (2023); Liu et al. (2024a). For example, FLIP Srivatsan et al. (2023) aligns image and text features through contrastive pre-training to enhance cross-domain robustness, while CFPL-FAS Liu et al. (2024a) generates semantic prompts from content and style features to dynamically modulate vision features. Compared with static tokens, S-CPTL Guo et al. (2024) further introduces dynamic prompts that adaptively capture instance-specific cues and increase diversity, thereby reducing overfitting.

**DeepFake Detection (DFD).** The primary goal of DFD is to counter digital attacks such as face swapping and expression manipulation, thereby safeguarding content authenticity. Early studies mainly exploited spatial-domain cues: some modeled global representations with CNNs, while others emphasized local receptive fields to detect forged patches Haliassos et al. (2021); Chai et al. (2020). To improve robustness, gradient-based features Ojha et al. (2023); Tan et al. (2023), adversarial training He et al. (2021), and regularization techniques Chen et al. (2022) have been explored. Frequency artifacts have also proven highly effective Frank et al. (2020); Durall et al. (2020), motivating approaches that leverage color space transformations, spectral discrepancies, or universal high-frequency modeling to boost cross-domain generalization Masi et al. (2020); Qian et al. (2020); Luo et al. (2021). More recently, with the advance of multimodal models and contrastive learning, prompt-based fine-tuning strategies have been proposed to exploit multimodal priors for deepfake detection Guo et al. (2025); Tan et al. (2025); Lin et al. (2025); Cui et al. (2025); Miao et al. (2025). In parallel, interpretability studies seek to uncover model reasoning, mitigate bias, and ensure ethical, regulation-compliant decisions Lo et al. (2025); Xu et al. (2024); Huang et al. (2025); Jia et al. (2024).

**Unified Face Attack Detection (UAD).** UAD seeks a universal model capable of handling both spoofing and deepfake attacks Yu et al. (2024); Deb et al. (2023); Fang et al. (2024); Chen et al. (2025); Liu et al. (2025). On the data side, JFSFDB Yu et al. (2024) integrates FAS and DFD datasets into the first joint benchmark, while UniAttackData Fang et al. (2024) introduces identity-consistent face-swapping samples to reduce domain noise. UniAttackData+ Liu et al. (2025) further incorporates diffusion-based attacks, enhancing diversity and difficulty. On the algorithmic side, JFSFDB employs a dual-branch physiological network, UniAttackData leverages a vision–language model with teacher–student prompting, MoAE-CR Chen et al. (2025) applies mixture-of-experts and distillation, and HiPTune Liu et al. (2025) adaptively integrates semantic cues through dynamic interactions. Motivated by these studies, we build upon CLIP and address classification ambiguity and spoofing diversity through a fine-tuning strategy tailored for UAD.

## 3 PRELIMINARIES: CONTRASTIVE LANGUAGE-IMAGE PRE-TRAINING (CLIP)

CLIP (Radford et al., 2021) is a vision–language model pretrained on large-scale image–text pairs to produce a unified representation for an input image $\boldsymbol{I} \in \mathbb{R}^{\mathrm{H} \times \mathrm{W} \times 3}$ and its textual description.

In the vision branch, the image $\boldsymbol{I}$ is first split into $n$ fixed-size patches and linearly projected to the initial patch embeddings $\boldsymbol{E}_0 \in \mathbb{R}^{n \times d_v}$, where $d_v=768$ denotes the visual token embedding dimension. Let the $i$-th vision transformer block be $\mathcal{V}_i(\cdot)$, where $i \in \{1, 2, ..., K\}$, and a learnable class token $\boldsymbol{c}_{i-1} \in \mathbb{R}^{d_v}$ is prepended at the patch embeddings $\boldsymbol{E}_{i-1}$ to form the $i$-th layer visual input embedding tokens $\boldsymbol{Z}_v^{i-1} = [\boldsymbol{c}_{i-1}, \boldsymbol{E}_{i-1}]$. The layer-wise update is formulated as $\boldsymbol{Z}_v^i = [\boldsymbol{c}_i, \boldsymbol{E}_i] = \mathcal{V}_i(\boldsymbol{Z}_v^{i-1})$. The class token $\boldsymbol{c}_K$ of the last layer is projected to the shared V–L embedding space by an image projection layer to obtain the final visual representation $\boldsymbol{v} = \text{ImageProj}(\boldsymbol{c}_K) \in \mathbb{R}^{d_{vt}}$, where $d_{vt} = 512$ denotes the dimensionality of the shared V–L embedding space.

In the language branch, the template words are tokenized into the initial word embeddings $\boldsymbol{W}_0 = [\boldsymbol{w}_0^1, \boldsymbol{w}_0^2, \ldots, \boldsymbol{w}_0^m] \in \mathbb{R}^{m \times d_t}$, where $m$ is the length of text tokens and $d_t=512$ is the text embedding dimension. Let the $i$-th text transformer block be $\mathcal{T}_i(\cdot)$, where $i \in \{1, 2, ..., K\}$, and the layer-wise

update is denoted as $\boldsymbol{W}_i = \mathcal{T}_i(\boldsymbol{W}_{i-1})$. The text representation is taken from the last token at the final layer and projected to the shared space by a text projection layer: $\boldsymbol{t} = \text{TextProj}(\boldsymbol{w}_K^m) \in \mathbb{R}^{d_{vt}}$.

During training (fine-tuning), the model employs a set of linear classifiers corresponding to different class labels $\boldsymbol{y} \in \{1, 2, ..., C\}$, where $C$ is the total number of categories, and the template prompts are formed as "a photo of a $\langle \text{CLASS} \rangle$". For the image $\boldsymbol{I}$ with the label $\hat{\boldsymbol{y}}$ in downstream data $\mathcal{D}$, the model is optimized by minimizing the cross-entropy loss:

$$\mathcal{L}_{ce} = \min_{\Theta} \; \mathbb{E}_{(\boldsymbol{I}, \hat{\boldsymbol{y}}) \sim \mathcal{D}} \left[ -\log \frac{\exp\left(\text{sim}(\boldsymbol{v}, \boldsymbol{t}_{\hat{\boldsymbol{y}}})/\tau\right)}{\sum_{\boldsymbol{y} \in C} \exp\left(\text{sim}(\boldsymbol{v}, \boldsymbol{t}_{\boldsymbol{y}})/\tau\right)} \right], \tag{1}$$

where $\Theta$ denotes the learnable model parameters, $\boldsymbol{t}_{\boldsymbol{y}}$ presents the text representation of the class $\boldsymbol{y}$, and $\tau$ is a temperature parameter.

## 4 METHODS

### 4.1 CROSS-MODAL PROMPT TUNING (CMPT)

To align the language and vision representations, Khattak et al. (2023) proposes Multimodal Prompt Tuning (MaPLe) to enhance CLIP by injecting a set of learnable tokens into the text branch at every transformer block and mapping these tokens into the vision branch as visual prompts. Concretely, the input embeddings tokens of text transformer block $i+1$ are extended with learnable tokens $\boldsymbol{P}_{t,i} = \{\boldsymbol{p}_i^j\}_{j=1}^a$, where $a$ is the length of learnable tokens. In the language branch, the input text embedding tokens are described as $\boldsymbol{Z}_{t,i} = [\boldsymbol{P}_{t,i}, \boldsymbol{W}_{t,i}] = [\boldsymbol{p}_i^1, \boldsymbol{p}_i^2, \ldots, \boldsymbol{p}_i^a, \boldsymbol{w}_i^{a+1}, \boldsymbol{w}_i^{a+2}, \ldots, \boldsymbol{w}_i^m] \in \mathbb{R}^{m \times d_t}$, and the output of the $i$-th text transformer block is updated as $\boldsymbol{Z}_{t,i+1} = \mathcal{T}_i(\boldsymbol{Z}_{t,i})$. In the vision branch, a coupling function $\mathcal{E}(\cdot)$ is utilized to inject categorical semantics distilled from learnable tokens to strengthen image representation. The visual input embedding tokens of the $i$-th vision transformer block are augmented as $\boldsymbol{Z}_{v,i} = [\boldsymbol{c}_i, \boldsymbol{E}_i, \boldsymbol{P}_{v,i}] \in \mathbb{R}^{(1+n+b) \times d_v}$, where $\boldsymbol{P}_{v,i} = \mathcal{E}_i(\boldsymbol{P}_{t,i}) \in \mathbb{R}^{b \times d_v}$ is the visual prompts and $b$ denotes the length of prompt tokens. The output of the $i$-th vision transformer block is updated as $\boldsymbol{Z}_{v,i+1} = \mathcal{V}_i(\boldsymbol{Z}_{v,i})$.

However, the supervisory signal in MaPLe is unidirectional (from text to vision): the textual representations are not grounded or updated based on visual evidence. Consequently, the inherent modality gaps between language and vision representations persist. To resolve this issue, we propose Cross-Modal Prompt Tuning (CMPT) to enable bidirectional, layer-wise alignment by inserting a pair of cross-modal prompts at every transformer block. Briefly, we establish two distinct coupling functions $\mathcal{E}_{v,i}(\cdot)$ and $\mathcal{E}_{t,i}(\cdot)$ to generate cross-modal prompts for vision and language branches, respectively. The function $\mathcal{E}_{v,i}$ (Sec. 4.2) fuses semantically related word embeddings for each class label and projects them as visual prompts, while $\mathcal{E}_{t,i}$ (Sec. 4.3) extracts fine-grained facial attack cues from patch embeddings to enhance the learnable tokens in the language branch. For the input of the $i$-th layer, the vision input embeddings are augmented as $\boldsymbol{Z}_{v,i} = [\boldsymbol{c}_i, \boldsymbol{E}_i, \mathcal{E}_{v,i}(\boldsymbol{w}_i^m)] \in \mathbb{R}^{(1+n+b) \times d_v}$, and the text input embeddings are denoted as $\boldsymbol{Z}_{t,i} = [\boldsymbol{P}_{t,i} + \mathcal{E}_{t,i}(\boldsymbol{E}_i), \boldsymbol{W}_i] \in \mathbb{R}^{m \times d_t}$.

### 4.2 SYNONYM SEMANTIC AUGMENTATION (SSA)

Due to the textual representation of labels being semantically coarse, SSA is introduced to enrich each category with context-adaptive synonym copies. Specifically, we generate a visual prompt for each class and concatenate them with the original visual embeddings to enhance the visual representation. For a given label $\boldsymbol{y}$, we extract the last embedding token $\boldsymbol{w}_{i,\boldsymbol{y}}^m$ from the text embeddings $\boldsymbol{Z}_{t,i}^{\boldsymbol{y}}$, and augment it by incorporating similar word embeddings retrieved from the vocabulary $\mathcal{X} \in \mathbb{R}^{e \times d_t}$, where $e$ denotes the vocabulary size. We first construct a query vector $\boldsymbol{q}_{i,\boldsymbol{y}} = \psi_1^i(\boldsymbol{w}_{i,\boldsymbol{y}}^m) \in \mathbb{R}^{d_t}$, where $\psi_1^i(\cdot)$ is a lightweight MLP transformation. The query vector is then compared against the vocabulary embeddings using cosine similarity, and the top $h$ most semantically similar tokens are selected. The similarity score is formulated as $\boldsymbol{S}_{\boldsymbol{y}}^i = \text{softmax}(\text{top}_h(\boldsymbol{q}_{\boldsymbol{y}}^i \cdot \mathcal{X}^\top)) = \{\boldsymbol{s}_{1,\boldsymbol{y}}^i, \boldsymbol{s}_{2,\boldsymbol{y}}^i, \ldots, \boldsymbol{s}_{h,\boldsymbol{y}}^i\} \in \mathbb{R}^h$, and $\mathcal{X}_h^i = \{\boldsymbol{x}_1^i, \boldsymbol{x}_2^i, \ldots, \boldsymbol{x}_h^i\} \in \mathbb{R}^{h \times d_t}$ stacks the top-$h$ synonym embeddings selected from vocabulary at $i$-th layer, where each $\boldsymbol{s}_{j,\boldsymbol{y}}^i$ corresponds to the softmax weight assigned to the $j$-th synonym candidate,

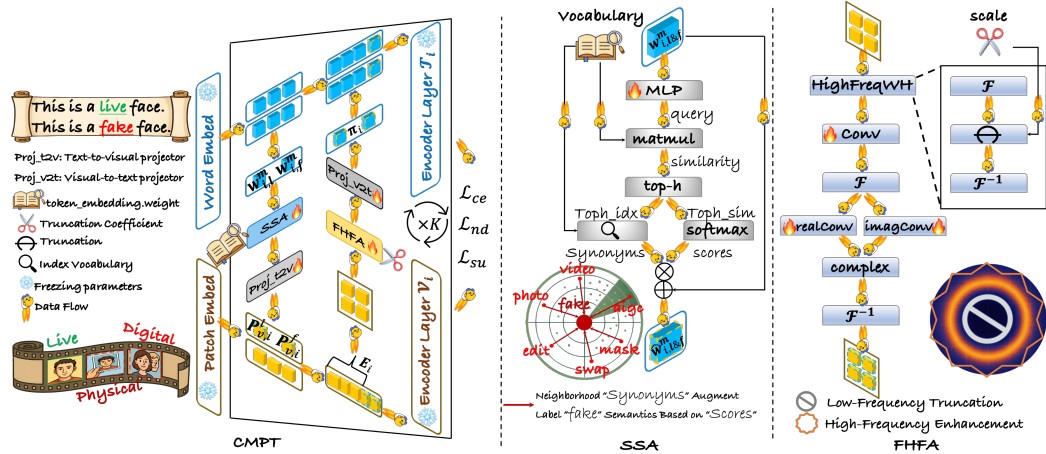

Figure 2: Overview of the proposed CMPT. The language branch employs SSA to retrieve top-$h$ semantic neighbors from a frozen vocabulary and aggregate them into semantically augmented category copies, which are projected (t2v) and injected as category prompts into the vision encoder. The vision branch uses an FHFA to suppress low frequencies and enhance high-frequency real/imaginary components via learnable convolutions, producing instance-conditioned visual prompts that are projected (v2t) into the language encoder. Bidirectional prompt transfer realigns vision and language while encoders remain frozen; only SSA/FHFA and projection layers are learnable.

and $\boldsymbol{x}_j^i$ denotes the synonym embedding. The augmented embedding token is defined as:

$$\widehat{\boldsymbol{w}}_{i,\boldsymbol{y}}^m = \boldsymbol{w}_{i,\boldsymbol{y}}^m + \sum_{j=1}^h (\boldsymbol{s}_{j,\boldsymbol{y}}^i \cdot \boldsymbol{x}_j^i). \tag{2}$$

Finally, the augmented embedding token $\widehat{\boldsymbol{w}}_{i,y}^m \in \mathbb{R}^{d_t}$ is projected into the vision space to obtain the class-specific visual prompt $\boldsymbol{P}_{v,i}^y = \mathrm{Proj}_{t2v}^i(\widehat{\boldsymbol{w}}_{i,y}^m) \in \mathbb{R}^{b \times d_v}$, where $\mathrm{Proj}_{t2v}^i(\cdot)$ is the $i$-th learnable linear projection layer, and the coupling function for visual prompts generation is defined as $\boldsymbol{P}_{v,i}^y = \mathcal{E}_{v,i}(\boldsymbol{w}_{i,y}^m)$. Notably, the visual prompts in the first visual transformer layer are derived from the input text template. Since the UAD task involves only two labels $\boldsymbol{y} \in \{live, fake\}$, the complete visual input embeddings of the $i$-th vision transformer block consist solely of the corresponding label-specific visual prompts $\boldsymbol{P}_{v,i}^l$ and $\boldsymbol{P}_{v,i}^f$. Formally, the visual input embeddings are defined as $\boldsymbol{Z}_{v,i} = [\boldsymbol{c}_i, \boldsymbol{E}_i, \boldsymbol{P}_{v,i}^l, \boldsymbol{P}_{v,i}^f] \in \mathbb{R}^{(1+n+2b) \times d_v}$.

### 4.3 FOURIER-BASED HIGH-FREQUENCY AMPLIFIER (FHFA)

To explore a unified discriminative space for both physical and digital attacks inspired by FreqNet (Tan et al., 2024), we employ a high-frequency filter to extract high-frequency cues and map them into learnable tokens as cross-modal biases that adaptively refine the text embeddings. Specifically, at the $i$-th layer, the FHFA module extracts the high-frequency components from the patch tokens $\boldsymbol{E}_i$. The extraction mask is defined as:

$$\mathcal{M} = \begin{cases} 1, & \text{if } |\mathrm{u}| > \alpha\mathrm{U}, |\mathrm{v}| > \alpha\mathrm{V}, \\ 0, & \text{otherwise,} \end{cases} \tag{3}$$

where $\mathcal{M} \in \mathbb{R}^{\mathrm{U} \times \mathrm{V}}$ is the high-frequency mask having the same spatial size as the patch tokens, U and V denote the height and width, and $\alpha$ is the ratio controlling the proportion of preserved high-frequency information. The masked frequency features are split into amplitude and phase spectra, denoted by $\boldsymbol{f}_{am}$ and $\boldsymbol{f}_{ph}$, respectively, such that

$$\boldsymbol{f}_{am} + \boldsymbol{f}_{ph}\mathrm{i} = (\mathcal{M} \cdot \mathcal{F}(\boldsymbol{E}_i)), \tag{4}$$

where $\mathcal{F}(\cdot)$ denotes the Fourier Frequency transform. To capture discriminative patterns, the extracted high-frequency features are processed through multiple convolutional blocks, and the final enhanced

patch tokens $\hat{\boldsymbol{E}}_i \in \mathbb{R}^{n \times d_v}$ are computed as:

$$\hat{\boldsymbol{E}}_i = \phi_3^i \left( \mathcal{F}^{-1} \left( \phi_1^i(\boldsymbol{f}_{am}) + \phi_2^i(\boldsymbol{f}_{ph}) \right) \right), \tag{5}$$

where $\mathcal{F}^{-1}(\cdot)$ is the inverse Fourier Frequency transform, and $\phi_1^i, \phi_2^i, \phi_3^i$ denote CNN blocks in $i$-th layer responsible for amplitude refinement, phase refinement, and final feature integration, respectively. The enhanced high-frequency tokens $\hat{\boldsymbol{E}}i$ are then projected into the text space via the $i$-th projection layer $\mathrm{Proj}_{v2t}^i(\cdot)$ to produce cross-modal biases $\boldsymbol{\pi}_i = \mathrm{Proj}_{v2t}^i(\hat{\boldsymbol{E}}_i) \in \mathbb{R}^{d_t}$. At the $i$-th layer, the overall cross-modal bias injection can be formulated as $\boldsymbol{\pi}_i = \mathcal{E}_{t,i}(E_i)$. It is worth noting that for the first layer, the patch tokens are derived from the original patch tokens as input.

Since the UAD task requires discriminating both real faces and attack types in the same feature space, we apply these biases to refine the attack-agnostic text embeddings. Specifically, the live text embeddings $\boldsymbol{Z}_{t,i}^l$ and unified fake text embeddings $\boldsymbol{Z}_{t,i}^f$ are updated as:

$$\boldsymbol{Z}_{t,i}^l = [\boldsymbol{P}_{t,i} + \boldsymbol{\pi}_i, \boldsymbol{W}_i^l] \in \mathbb{R}^{m \times d_t}, \quad \boldsymbol{Z}_{t,i}^f = [\boldsymbol{P}_{t,i} + \boldsymbol{\pi}_i, \boldsymbol{W}_i^f] \in \mathbb{R}^{m \times d_t}. \tag{6}$$

Here, the cross-modal bias serves as an auxiliary signal to guide the text branch toward capturing subtle forgery traces from the visual domain. By adaptively refining the learnable tokens, FHFA allows the text branch to adjust its decision boundaries according to the high-frequency cues extracted from the image, thereby strengthening the semantic alignment between the visual and textual modalities.

### 4.4 Loss Functions

**Synonym Uniformity Loss.** To prevent the synonym selection from collapsing onto a single candidate, we regularize the distribution of the synonym scores $\boldsymbol{S}_{\boldsymbol{y}}^i$ at each layer by enforcing it to be close to a uniform distribution over the top-$h$ neighbors. Formally,

$$\mathcal{L}_{su} = \frac{1}{|C|} \sum_{\boldsymbol{y} \in C} \sum_{i=1}^{K} D_{\mathrm{KL}}(\boldsymbol{S}_{\boldsymbol{y}}^i \parallel \boldsymbol{U}_h), \tag{7}$$

where $\boldsymbol{U}_h = [\frac{1}{h}, \dots, \frac{1}{h}] \in \mathbb{R}^h$ denotes the uniform distribution over the top-$h$ synonyms, and $K$ is the number of transformer layers. Minimizing this loss is equivalent to maximizing the entropy of $\boldsymbol{S}_{\boldsymbol{y}}^i$, thereby encouraging a more diverse and robust utilization of synonym candidates.

**Neighbor Diversity Loss.** To encourage the model to explore a diverse set of synonym candidates rather than selecting highly redundant neighbors, we introduce a neighbor diversity loss. Specifically, let the top-$h$ synonym embeddings selected at the $i$-th transformer layer be denoted as $\mathcal{X}_h^i \in \mathbb{R}^{h \times d_t}$, where $\boldsymbol{x}_j^i$ denotes the synonym embedding. The neighbor diversity loss is then defined as the mean of the pairwise similarities among the selected synonyms:

$$\mathcal{L}_{nd} = \frac{1}{K} \sum_{i=1}^{K} \frac{1}{h^2} \sum_{j=1}^{h} \sum_{j'=1}^{h} \langle \boldsymbol{x}_j^i, \boldsymbol{x}_{j'}^i \rangle. \tag{8}$$

Minimizing $\mathcal{L}_{nd}$ penalizes excessive similarity among the selected synonyms, thereby encouraging the model to select more diverse neighbors. This promotes richer semantic representations and reduces redundancy in the synonym space.

**Total Loss.** In this paper, we adopt the cross-entropy loss $\mathcal{L}_{ce}$ (defined in Eq. 1) as our primary objective. In addition, we introduce two auxiliary losses: the synonym uniformity loss $\mathcal{L}_{su}$, which prevents synonym scores from collapsing onto a single neighbor; and the neighbor diversity loss $\mathcal{L}_{nd}$, which discourages overly redundant neighbors. The total loss with the hyperparameters $\lambda_1$ and $\lambda_2$ is therefore formulated as

$$\mathcal{L}_{total} = \mathcal{L}_{ce} + \lambda_1 \mathcal{L}_{su} + \lambda_2 \mathcal{L}_{nd}. \tag{9}$$

## 5 Experiments

### 5.1 Experimental Setup

**Datasets, Protocols, and Evaluation Metrics.** We evaluate UAD-CMPT on two UAD benchmarks: *JFSFDB* (Yu et al., 2024) and *UniAttackData* (Fang et al., 2024). On *JFSFDB*, we conduct cross-domain evaluation under two settings: (i) separate training for FAS or DFD and (ii) joint training

Table 1: The results (%) of JFSFDB datasets. ↓/↑ represents that the smaller/larger value, the better performance. Best results are in **bold**.

| Methods | FAS | | DFD | | Uni-Attack | | Average | |
|---|---|---|---|---|---|---|---|---|
| | EER(%↓) | AUC(%↑) | EER(%) | AUC(%) | EER(%) | AUC(%) | EER(%) | AUC(%) |
| MesoNet (WIFS'18) | 38.18 | 65.97 | 42.47 | 59.91 | 42.11 | 61.10 | 40.92 | 62.33 |
| DeepPixel (IJCB'19) | 30.12 | 77.55 | 29.82 | 76.53 | 28.64 | 78.00 | 29.53 | 77.36 |
| CDCN++ (TPAMI'20) | 35.86 | 69.02 | 36.47 | 67.50 | 36.64 | 70.04 | 36.32 | 68.85 |
| MultiAtten (CVPR'21) | 37.87 | 66.25 | 40.10 | 63.86 | 35.21 | 69.36 | 37.73 | 66.49 |
| CLIP (ICML'21) | 18.07 | 89.70 | 25.15 | 82.74 | 22.35 | 85.32 | 21.86 | 85.92 |
| CoOp (IJCV'22) | 18.34 | 83.43 | 40.31 | 63.25 | 27.43 | 79.58 | 28.69 | 75.42 |
| ViT-shared8 (TDSC'24) | - | - | - | - | 22.26 | 85.26 | 22.26 | 85.26 |
| UAD-CMPT(Ours) | **10.02** | **95.60** | **21.27** | **86.98** | **20.57** | **87.78** | **17.29** | **90.12** |

for UAD. In the separate setting, FAS is trained on 3DMAD (Erdogmus & Marcel, 2014), SiW (Liu et al., 2018b), HKBU (Liu et al., 2016) and tested on 3DMask (Yu et al., 2020a), MSU (Wen et al., 2015), ROSE (Li et al., 2018), while DFD is trained on FF++ (Rossler et al., 2019) and tested on DFDC (Dolhansky et al., 2019), CelebDFv2 (Li et al., 2020); in the joint setting UAD, a single model is trained on SiW, 3DMAD, HKBU, FF++ and evaluated on MSU, 3DMask, ROSE, DFDC, CelebDFv2. For *UniAttackData*, Protocol 1 (P1) evaluates unified detection with all attack types present in both training and testing. Protocol 2 (P2) adopts a leave-one-type-out scheme to assess generalization to unseen attacks. We also report additional protocols: Protocol 1.1 (P1.1) and Protocol 1.2 (P1.2) exclude deepfake and adversarial attacks during training/validation and evaluate on disjoint identities, whereas Protocol 1.3 (P1.3) includes all digital subtypes under the standard distribution. We also evaluate on the DG benchmark for FAS, comprising four datasets, MSU-MFSD (M) (Wen et al., 2015), CASIA-FASD (C) (Zhang et al., 2012), Idiap Replay-Attack (I) (Chingovska et al., 2012), OULU-NPU (O) (Boulkenafet et al., 2017), treating each dataset as a distinct domain. We follow a DG protocol, where A&B→C denotes training on the union of A and B as source domains and evaluating on C as the unseen target.

We assess performance with three measures: (1) Average Classification Error Rate (ACER), computed as the mean of the false rejection rate (FRR) and false acceptance rate (FAR); (2) Area Under the Curve (AUC), a threshold-free summary of discriminability; (3) Equal Error Rate (EER), the error rate at the operating point where FRR equals FAR.

**Implementation Details.** Our UAD-CMPT is built on the CLIP (Radford et al., 2021), where the image encoder $\mathcal{V}(\cdot)$ is a ViT-B/16 and text encoder $\mathcal{T}(\cdot)$ is a Transformer, with $d_v = 768$, $d_t = 512$, and $d_{vt} = 512$. In our approach, SSA, FHFA, and two coupling functions of each layer, $\text{Proj}_{t2v}$ and $\text{Proj}_{v2t}$, are trainable, while the remaining parameters are frozen. Unless otherwise stated, we set the number of top-$h$ in SSA to 10. Based on a large number of experimental summaries, we set $\lambda_1$ and $\lambda_2$ to be 0.01. Following FreqNet (Tan et al., 2024), we set $\alpha = 0.25$ to control the preserved high-frequency ratio. All models are trained with SGD optimizer for 100 epochs (each epoch only accesses one frame from a video) with a batch size of 1 and an initial learning rate of 0.02, which is decayed by the cosine annealing scheduler. Training stops after 100 epochs or earlier if the loss plateaus.

## 5.2 Unified Face Attack Detection Results

On the JFSFDB (Yu et al., 2024) benchmark, we evaluate high-performing classical DFD methods MesoNet (Afchar et al., 2018), MultiAtten (Zhao et al., 2021) and FAS methods DeepPixel George & Marcel (2019), CDCNN++ (Yu et al., 2020a), multimodal methods CLIP (Radford et al., 2021), CoOp (Zhou et al., 2022b), and the SOTA UAD method ViT-shared8 (Yu et al., 2024). From Tab. 1, we observe two conclusions: (1) Our UAD-CMPT surpasses all competing methods under the settings of FAS and DFD. In terms of EER, it outperforms the runner-up method, CLIP, by 8.05% and 3.88%, respectively. UAD-CMPT surpasses CLIP chiefly by suppressing low-frequency content and amplifying high-frequency magnitude and phase. Forgery traces that are hard to perceive in raw images become salient in the high-frequency domain. By centering decisions on these shared high-frequency cues, UAD-CMPT forms a cross-domain decision space for authenticity, yielding more stable cross-dataset performance and less degradation in EER and AUC. (2) Under the UAD setting, UAD-CMPT surpasses ViT-shared8 by 1.69% and ultimately achieves an average EER of 17.29%. CMPT's gains over ViT-shared8 chiefly stem from SSA. Because live and fake are semantically ambiguous for VLMs, SSA retrieves semantically related tokens from a frozen vocabulary and

Table 2: The results (%) of UniAttackData datasets. Avg. represents the average ACER of P1, P1.1, P1.2, and P1.3. Best results are in **bold**.

| Methods | P1 | | P1.1 | | P1.2 | | P1.3 | | Avg. | P2 | |
|---|---|---|---|---|---|---|---|---|---|---|---|
| | ACER | AUC | ACER | AUC | ACER | AUC | ACER | AUC | | ACER | AUC |
| CDCN++ (TPAMI'20) | 1.40 | 99.52 | 12.32 | 93.89 | 16.34 | 93.34 | 4.41 | 97.68 | 8.62 | 34.33 | 77.46 |
| CLIP (ICML'21) | 1.02 | 99.47 | 14.81 | 86.74 | 5.36 | 99.17 | 2.45 | 97.92 | 5.91 | 24.26 | 87.34 |
| UniAttackD (IJCAI'24) | 0.52 | 99.96 | 11.73 | 98.81 | 1.70 | 99.85 | 4.67 | 99.13 | 4.66 | 22.42 | **91.97** |
| MoAE-CR (AAAI'25) | 0.37 | 99.97 | - | - | - | - | - | - | - | 15.13 | 92.07 |
| FA$^3$-CLIP (TIFS'25) | 0.36 | 99.75 | 9.57 | 97.78 | **1.43** | **99.85** | 2.30 | 99.19 | 3.42 | 18.81 | 88.59 |
| UAD-CMPT (Ours) | **0.34** | **99.97** | **5.23** | **98.88** | 3.11 | 99.45 | **2.15** | **99.59** | **2.71** | **14.63** | 89.28 |

Table 3: The results (%) of Protocol 1 on M, C, I, and O datasets. A & B → C denotes training on the union of A and B as source domains and evaluating on C as the unseen target.

| Methods | O&C&I→M | | O&M&I→C | | O&C&M→I | | I&C&M→O | | Avg. |
|---|---|---|---|---|---|---|---|---|---|
| | HTER ↓ | AUC ↑ | HTER | AUC | HTER | AUC | HTER | AUC | HTER |
| UDG-FAS (ICCV' 23) | 7.14 | 97.31 | 11.44 | 95.59 | 6.28 | 98.61 | 12.18 | 94.36 | 9.26 |
| IADG (CVPR' 23) | 5.41 | 98.19 | 8.70 | 96.44 | 10.62 | 94.50 | 8.86 | 97.14 | 8.39 |
| HPDR (CVPR' 24) | 4.58 | 96.02 | 11.30 | 94.42 | 11.26 | 92.49 | 9.93 | 95.26 | 9.26 |
| TTDG-V (CVPR' 24) | 4.16 | 98.48 | 7.59 | 98.18 | 9.62 | 98.18 | 10.00 | 96.15 | 7.84 |
| CA-MoEiT (IJCV' 24) | 2.88 | 98.76 | 7.89 | 97.70 | 6.18 | 98.94 | 9.72 | 96.22 | 6.67 |
| GAC-FAS (CVPR' 24) | 5.00 | 97.56 | 8.20 | 95.16 | 4.29 | 98.87 | 8.60 | 97.16 | 6.52 |
| ViT-S-Adapter (TIFS' 24) | 2.90 | 99.48 | 7.37 | 97.63 | 8.54 | 97.17 | 8.20 | 97.69 | 6.74 |
| CFPL-FAS (CVPR' 24) | 3.09 | 99.45 | 2.56 | 99.10 | 5.43 | 98.41 | 3.33 | 99.05 | 3.60 |
| DCRN (TIFS' 25) | 4.05 | 99.12 | 7.38 | 97.57 | 6.17 | 98.22 | 8.33 | 98.17 | 6.48 |
| AG-FAS (TPAMI' 25) | 5.71 | 98.03 | 5.44 | 98.55 | 6.71 | 98.23 | 9.43 | 96.62 | 6.82 |
| FSFM (CVPR' 25) | 3.78 | 99.15 | 3.16 | 99.41 | 4.63 | 99.03 | 7.68 | 97.11 | 4.81 |
| OTA (CVPR' 25) | 2.38 | 99.42 | 2.67 | **99.49** | 5.19 | 98.56 | 3.03 | 99.45 | 2.91 |
| UAD-CMPT (Ours) | **0.71** | **99.81** | **1.66** | 98.96 | **4.28** | **99.19** | **2.22** | **99.65** | **2.21** |

aggregates them to enrich class prompts, aligning language representations with diverse physical and digital forgeries. This reduces categorical ambiguity and strengthens unified attack detection.

On the UniAttackData (Fang et al., 2024) benchmark, we select CDCNN++, CLIP, and three recently proposed UAD algorithms, UniAttackD (Fang et al., 2024), MoAE-CR (Chen et al., 2025), and FA$^3$-CLIP (Li et al., 2025a) for experiments. Except for P1.2, UAD-CMPT achieves the best performance across all other protocols, with particularly significant gains on protocol P1.1 and P2, where its ACER substantially surpasses that of the second-best algorithm FA$^3$-CLIP (9.57% vs. 5.23% for P1.1 and 18.81% vs. 14.63% for P2). According to the definitions, P1.2 excludes adversarial attacks from training and evaluates on disjoint identities. Without adversarial samples, UAD-CMPT's frequency-centric bias is disadvantaged: adversarial perturbations are subtle and only weakly represented in the high-frequency spectrum, so the model's high-frequency emphasis yields less benefit.

## 5.3 DOMAIN GENERALIZATION RESULTS

We also compare UAD-CMPT with some of the currently optimal DG algorithms, including vision-only modal algorithms (i.e., AG-FAS (Long et al., 2024), OTA (Li et al., 2025b) and FSFM (Wang et al., 2025)), multimodal algorithms (i.e., CFPL-FAS (Liu et al., 2024a)). As shown in Tab. 3, UAD-CMPT consistently achieves the lowest HTER across all cross-domain settings. In particular, it records only 0.71% on O&C&I→M and 2.22% on I&C&M→O, significantly outperforming previous best methods such as OTA, FSFM and CFPL-FAS. On average, UAD-CMPT attains an HTER of 2.21%, establishing a new state-of-the-art and demonstrating superior cross-domain generalization. These results validate that the proposed bidirectional prompt-transfer design not only benefits unified attack detection but also substantially enhances domain generalization performance.

## 5.4 ABLATION STUDY

**Contribution of Each Component.** To investigate the contribution of each improvement in UAD-CMPT, such as SSA and FHFA, we gradually introduce them on the baseline IVLP (Khattak et al., 2023) and report the ACER results of UniAttackData (Fang et al., 2024) in Tab. 4. Starting from the naive baseline IVLP (Khattak et al., 2023), which simply combines vision and language prompts, we observe clear performance gains when introducing SSA and FHFA. Specifically, SSA enriches the

Table 4: The effect of SSA and FHFA. ↓ represents the performance benefit compared to IVLP.

| Methods | | | P1 | P1.1 | P1.2 | P1.3 | Avg. |
|---|---|---|---|---|---|---|---|
| IVLP | SSA | FHFA | | | | | |
| ✓ | ✗ | ✗ | 0.88 | 8.06 | 7.89 | 2.94 | 4.94 |
| ✓ | ✓ | ✗ | 0.48 | 5.66 | 3.25 | 2.28 | 2.91 |
| ✓ | ✗ | ✓ | 0.65 | 5.88 | 5.34 | 2.63 | 3.62 |
| ✓ | ✓ | ✓ | **0.34** | **5.23** | **3.11** | **2.15** | **2.71** |
| | | | (↓0.54) | (↓2.83) | (↓4.78) | (↓0.79) | (↓2.23) |

Table 5: Effect of $\mathcal{L}_{ce}$, $\mathcal{L}_{su}$, and $\mathcal{L}_{nd}$. ↓ represents the performance benefit compared to $\mathcal{L}_{ce}$.

| Methods | | | P1 | P1.1 | P1.2 | P1.3 | Avg. |
|---|---|---|---|---|---|---|---|
| $\mathcal{L}_{ce}$ | $\mathcal{L}_{su}$ | $\mathcal{L}_{nd}$ | | | | | |
| ✓ | ✗ | ✗ | 0.65 | 7.38 | 4.49 | 3.64 | 4.04 |
| ✓ | ✓ | ✗ | 0.60 | 7.12 | 3.25 | 3.25 | 3.55 |
| ✓ | ✗ | ✓ | 0.41 | 6.36 | 3.11 | 2.77 | 3.16 |
| ✓ | ✓ | ✓ | **0.34** | **5.23** | **3.11** | **2.15** | **2.71** |
| | | | (↓0.31) | (↓2.15) | (↓1.38) | (↓1.49) | (↓1.33) |

Figure 3: The UMAP (McInnes et al., 2018) projection of UAD-CMPT's penultimate layer on UniAttackData. Points are colored by attack subtype; markers denote class (○ live, + fake).

class descriptions by aggregating semantically related tokens, thereby alleviating the ambiguity of the live/fake labels and providing more precise semantic guidance for distinguishing diverse physical and digital forgeries. In parallel, FHFA highlights high-frequency amplitude and phase cues while suppressing low-frequency content, enabling the model to focus on forgery artifacts that are more stable across attack types. When integrated, SSA and FHFA complement each other and yield the best overall results, reducing the average ACER from 4.94% to 2.71%.

**Contribution of Each Constraint.** Tab. 5 presents the ablation study on different loss configurations. Using only the cross-entropy loss $\mathcal{L}_{ce}$ yields the weakest performance, with an average ACER of 4.04%. Introducing the uniformity loss $\mathcal{L}_{su}$ improves the results to 3.55%, indicating that encouraging a more balanced distribution of retrieved synonyms prevents the model from collapsing onto a few dominant prompts. Replacing $\mathcal{L}_{su}$ with the neighbor diversity loss $\mathcal{L}_{nd}$ further reduces the average ACER to 3.16%, showing the benefit of enforcing diversity among neighboring prompts. When combining all three objectives, the model achieves the best overall performance with an average ACER of 2.71%, a relative reduction of 1.33% compared to the baseline. These results highlight that $\mathcal{L}_{su}$ and $\mathcal{L}_{nd}$ play complementary roles: the former regularizes the distribution of semantic augmentations, while the latter enhances their diversity, and together they yield more robust and discriminative representations.

## 5.5 Visualization and Analysis

As shown in Fig. 3, for Protocols P1, P1.1, and P1.3, our UAD-CMPT separates live faces from all forgery types with clear margins. However, under P1.2, the live–fake decision boundary becomes noticeably less distinct. We attribute this to FHFA biasing the model toward spectral cues that are weak or absent for several attack types. Adversarial perturbations are designed to be imperceptible and seldom yield strong high-frequency signatures, while structural signals, such as printed-photo or screen borders and global quality variations, are predominantly low-frequency and global; consequently, suppressing low frequencies can remove the very evidence needed to detect these attacks.

## 6 Conclusion

In this work, we introduced UAD-CMPT, a cross-modal prompt-tuning framework that addresses categorical ambiguity and forgery diversity in unified face attack detection. By integrating SSA for enriched semantic prompts and FHFA for robust spectral cues, UAD-CMPT effectively restores vision–language alignment and establishes a shared discriminative space.

## REPRODUCIBILITY STATEMENT

We provide model details, training setup, and data preprocessing in the main text, and will release anonymized source code with scripts for data downloading/preparation, training, and evaluation. Exact configuration files, environment specifications, fixed random seeds, dataset splits, and metric definitions are included to enable step-by-step replication. For SSA and FHFA we adopt stable defaults: top-$h = 10$ and $\alpha = 0.25$, while acknowledging that these hyperparameters *materially influence* performance and are not universally optimal across benchmarks and protocols with different forgery types and visual characteristics. To support both exact reproduction and adaptation, we provide per-benchmark configuration files and short sweep scripts, and recommend limited retuning within small ranges (e.g., top-$h \in \{5, 10, 15\}$ and $\alpha \in \{0.15, 0.25, 0.35\}$).

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
