# OpenReview forum: "UAD-CMPT: Unified Face Attack Detection via Cross-Modal Prompt Tuning"
_ICLR.cc/2026/Conference — ICLR 2026 Conference Withdrawn Submission_

### Official Review · Reviewer_HnzG · 2025-10-24

**Soundness:** 2
**Presentation:** 3
**Contribution:** 2
**Rating:** 4
**Confidence:** 4

**Summary:**

This paper presents UAD-CMPT, a cross-modal prompt-tuning framework for unified face attack detection that integrates both physical and digital forgery detection. It tackles categorical ambiguity and forgery diversity by combining two key modules: Synonym Semantic Augmentation (SSA), which enriches textual labels with semantic neighbors, and Fourier-based High-Frequency Amplifier (FHFA), which enhances discriminative high-frequency visual cues. These modules enable bidirectional alignment between vision and language. Experiments on major benchmarks show state-of-the-art performance and improved generalization of the proposed method.

**Strengths:**

1.	The paper addresses the problems of categorical ambiguity and forgery diversity in unified face attack detection and proposes a bidirectional prompt-transfer mechanism to tackle these challenges simultaneously.

2.	UAD-CMPT achieves SOTA performance across multiple UAD protocols, with substantial gains on cross-domain settings.

3.	The ablation studies and UMAP visualization demonstrate the effects of SSA and FHFA.

**Weaknesses:**

Major Weaknesses

1.	FHFA depends heavily on the high-frequency ratio α. The paper acknowledges this but provides no sensitivity analysis or justification
for choosing this value across datasets.

2.	The process of retrieving and aggregating synonym embeddings from the frozen vocabulary lacks detail. How are these vocabularies chosen and are they domain-specific to this task?

3.	The ablation studies cover SSA and FHFA but do not include comparisons with other cross-modal alignment baselines (e.g., MaPLe).

4.	The training setup uses frozen encoders and small batch size (1 frame/video), which may not be suitable for real-world deployment. In addition, training and inference computational cost and hardware requirements are expected.

5.	The model faces degradation under Protocol P1.2, yet the discussion of the reason why FHFA harms performance is brief.

Minor Weaknesses

1.	The paper could benefit from a more detailed qualitative comparison (e.g., visual heatmaps) to show how SSA changes category semantics.

**Questions:**

1. The authors should provide a sensitivity analysis of the high-frequency ratio and justify the principle of chosen values.

2. The authors need to clarify how the frozen vocabulary for SSA is built, specify whether it is domain-specific. Does this paradigm introduce potential impact on semantic bias?

3. Comparisons with cross-modal alignment baselines such as MaPLe are expected to demonstrate the contribution of UAD-CMPT.

4. What is the training and inference costs and hardware settings?

5. An analysis of performance degradation under Protocol P1.2 benefits the contributions of the work, including the possible reasons and potential improvements of future work.

6. The authors could additionally include qualitative visualizations such as attention maps to illustrate how SSA and FHFA enhance semantic and visual alignment.

---

### Official Review · Reviewer_uof8 · 2025-10-26

**Soundness:** 4
**Presentation:** 4
**Contribution:** 3
**Rating:** 6
**Confidence:** 4

**Summary:**

The paper proposes UAD-CMPT, a unified framework for face attack detection that leverages cross-modal prompt tuning on pre-trained vision-language models like CLIP to address categorical ambiguity in labels ("live" vs. "fake") and diversity in forgery cues from physical and digital attacks. Key contributions include Synonym Semantic Augmentation (SSA) for enriching textual semantics via synonym retrieval and weighted aggregation, and a Fourier-based High-Frequency Amplifier (FHFA) for consolidating visual forgery cues through adaptive high-frequency enhancement. This bidirectional prompt transfer realigns vision and language modalities. Experiments on benchmarks such as JFSFDB, UniAttackData, and cross-domain settings show superior performance in metrics like EER, AUC, and ACER compared to state-of-the-art FAS, DFD, and UAD methods.

**Strengths:**

Originality: Introduces a novel bidirectional prompt-tuning framework (CMPT) tailored for UAD, creatively combining SSA for language-side semantic expansion with FHFA for vision-side frequency-based cue consolidation, extending prior prompt engineering in VLMs to handle task-specific challenges like categorical ambiguity and forgery heterogeneity.
Quality: Provides a solid experimental validation, including intra- and cross-domain evaluations on multiple UAD benchmarks, comparisons with baselines (e.g., CLIP, CoOp, and UAD-specific methods like HiPTune), and ablations on components (SSA, FHFA) and losses, demonstrating consistent improvements.
Clarity: Well-organized structure with clear explanations of challenges, framework, and modules; informative figures and contextualization of related work in FAS, DFD, and UAD enhance readability.
Significance: Advances unified detection for both physical and digital face attacks, potentially reducing model deployment costs in face recognition systems and showcasing VLMs' applicability in security domains.

**Weaknesses:**

Over-reliance on High-Frequency Cues: The method's design heavily depends on the assumption that high-frequency artifacts serve as the primary and most generalizable clues across all forgery types. While effective for many digital attacks that introduce high-frequency noise, this assumption may not hold for certain physical attacks.
Limited Novelty in Components: Although innovative overall, FHFA largely builds on existing frequency-domain techniques in DFD (e.g., Frank et al., 2020; Luo et al., 2021); the paper could better highlight distinctions in its adaptive convolutions for real/imaginary parts to emphasize uniqueness.
Missing Hyperparameter Analysis: The method introduces two important hyperparameters: $top-h$ (for SSA) and $\alpha$ (for FHFA's high-frequency ratio). The reproducibility statement mentions that these "significantly affect performance" and provides scanning ranges, but the main paper does not offer sensitivity analysis or rationale for the selected default values. Demonstrating ablation studies on performance variations with these values would enhance the paper's completeness.

**Questions:**

Given the insightful analysis of the performance drop on protocol P1.2—namely, that FHFA suppresses low-frequency cues useful for certain attack types—have you considered modifying FHFA? For example, could a module be designed to learn which frequency bands to amplify, rather than relying on a fixed high-pass filter?
Could you provide more details on the frozen vocabulary used in SSA? How was it constructed, and what is its size? Clarifying this could address potential reproducibility concerns.
Since FHFA is inspired by FreqNet, could you elaborate on its distinctions? For instance, how do the adaptive convolutions on real/imaginary parts in FHFA provide advantages over FreqNet's amplitude/phase processing, particularly for unifying physical and digital attacks in UAD?

---

### Official Review · Reviewer_oUzN · 2025-10-29

**Soundness:** 3
**Presentation:** 2
**Contribution:** 3
**Rating:** 4
**Confidence:** 3

**Summary:**

This paper proposes UAD-CMPT, a cross-modal prompt tuning framework for unified face attack detection that addresses categorical ambiguity through Synonym Semantic Augmentation (SSA) and consolidates diverse forgery cues via a Fourier-based High-Frequency Amplifier (FHFA). The bidirectional prompt transfer design is innovative, and the method achieves competitive or state-of-the-art results on multiple UAD benchmarks including JFSFDB and UniAttackData.

**Strengths:**

1. The paper identifies an important practical challenge—unified detection of diverse physical and digital face attacks—and clearly articulates two key bottlenecks for VLMs: categorical ambiguity of live/fake labels and heterogeneous forgery cues across attack types.
2. Unlike prior unidirectional methods (e.g., MaPLe), CMPT introduces genuine bidirectional alignment where SSA enriches semantics (language→vision) and FHFA extracts discriminative cues (vision→language), creating a theoretically appealing complementary framework.
3. The paper evaluates across multiple benchmarks (JFSFDB, UniAttackData with 5 protocols, 4-dataset domain generalization) and demonstrates competitive or state-of-the-art performance in most settings, with particularly strong results on cross-domain generalization.

**Weaknesses:**

1. Both key claims—that CLIP suffers from "categorical ambiguity" for live/fake labels and that high-frequency features provide a "unified discriminative space"—lack direct empirical evidence. The paper provides no visualization of retrieved synonyms, no spectral analysis comparing forgery types, and no attention maps demonstrating CLIP's misalignment. Additionally, the "frozen vocabulary" source, scale, and semantic domain are not specified anywhere (including appendix), despite critically affecting retrieval results and semantic space.

2. Critical ablations are missing (sensitivity to α and top-h values), statistical significance is not reported (no error bars or confidence intervals), and key baseline MoAE-CR has incomplete results (only P1 and P2 in Table 2). The paper also omits computational cost analysis (training time, inference speed, parameter count) entirely.

3. Beyond UMAP projections showing class separation, the paper lacks: (1) concrete examples of retrieved synonyms and their quantitative impact on different attack types, (2) frequency-domain visualizations showing amplified patterns in amplitude/phase components, and (3) spatial attention maps (e.g., Grad-CAM-style heatmaps) highlighting discriminative frequency regions for individual samples.

4. Presentation Issues: Figure 2, while visually appealing, is overly complex and difficult to parse—the bidirectional arrows, multiple projection layers, and concurrent SSA/FHFA operations are not clearly delineated, hindering reader comprehension of the actual information flow.

**Questions:**

1. The proposed Fourier-based High-Frequency Amplifier (FHFA) explicitly suppresses low-frequency information, yet certain physical or adversarial attacks exhibit low-frequency or global cues. How does the model maintain robustness when such cues are dominant, and can the authors provide results on adaptive or low-pass attacks to validate the generality of FHFA?

2. The Synonym Semantic Augmentation (SSA) module retrieves top-h neighbors from a frozen vocabulary to expand label semantics. Could the authors clarify the vocabulary source, its domain overlap with training data, and the model’s sensitivity to different vocabularies or values of h? Is there any risk of semantic leakage or instability introduced by this retrieval process?

3. The proposed CMPT performs layer-wise bidirectional prompt exchange between vision and language encoders, but the paper does not analyze training stability, convergence behavior, or computational overhead. Can the authors provide ablations on which layers contribute most, and report FLOPs or runtime comparisons against CLIP or MaPLe baselines?

---

### Official Review · Reviewer_8ecX · 2025-10-31

**Soundness:** 3
**Presentation:** 2
**Contribution:** 2
**Rating:** 2
**Confidence:** 4

**Summary:**

This article proposes a bidirectional vision-text supervision framework to jointly address attacks in both physical and digital domains. It introduces frequency-domain design in the visual branch and synonym semantic augmentation in the textual branch to enhance performance.

**Strengths:**

1.The motivation of this paper is strong, and it defines the key challenges of multi-task forgery detection.

2.The method is relatively clear and easy to follow.

3.The performance is good on multiple benchmarks.

**Weaknesses:**

1.Frequency-domain design has been proposed in FreqNet, and synonym augmentation has been introduced in FLIP; this article appears to be a combination of FreqNet and FLIP.

2.The method and presentation of synonym augmentation lack detail, making it difficult to discern the uniqueness of the design, and the corresponding ablation study is also insufficient. Therefore, the challenge of jointly addressing the two types of attacks has not been proven to be resolved.

3.The fonts in the figures of the article are messy and hard to read.

[1] Tan C, Zhao Y, Wei S, et al. Frequency-aware deepfake detection: Improving generalizability through frequency space domain learning[C]//Proceedings of the AAAI Conference on Artificial Intelligence. 2024, 38(5): 5052-5060.
[2] Srivatsan K, Naseer M, Nandakumar K. Flip: Cross-domain face anti-spoofing with language guidance[C]//Proceedings of the IEEE/CVF international conference on computer vision. 2023: 19685-19696.

**Questions:**

See weaknesses

---

### Note · Authors · 2025-11-14

I have read and agree with the venue's withdrawal policy on behalf of myself and my co-authors.